# Case of Early-Onset Parkinson’s Disease in a Heterozygous Mutation Carrier of the *ATP7B* Gene

**DOI:** 10.3390/jpm9030041

**Published:** 2019-08-17

**Authors:** Ekaterina Y. Ilyechova, Irina V. Miliukhina, Marina N. Karpenko, Iurii A. Orlov, Ludmila V. Puchkova, Sergey A. Samsonov

**Affiliations:** 1International Research Laboratory of Trace Elements Metabolism, ITMO University, Kronverksky av., 49, St. Petersburg 197101, Russia; 2Department of Molecular Genetics, Institute of Experimental Medicine, Pavlov str., 12, St. Petersburg 197376, Russia; 3Biophysics Department, Peter the Great St. Petersburg Polytechnic University, Politehknicheskay str., 29, St. Petersburg 195251, Russia; 4Centre for Neurodegenerative diseases, Institute of Experimental Medicine, Maluy av., Petrogradskiy district, 13, St. Petersburg 197198, Russia; 5Department of Physiology, Institute of Experimental Medicine, Pavlov str., 12, St. Petersburg 197376, Russia; 6Faculty of Chemistry, University of Gdańsk, Wita Stwosza str., 63, 80-308 Gdańsk, Poland

**Keywords:** Parkinson’s disease, Wilson’s disease, copper-status index, N-domain of ATP7B, molecular modeling

## Abstract

In this paper, we report a clinically proven case of Parkinson’s disease (PD) with early onset in a patient who is a heterozygous mutation carrier of *ATP7B* (the Wilson’s disease gene). The patient was observed from 2011 to 2018 in the Center for Neurodegenerative Diseases, Institute of Experimental Medicine (St. Petersburg, Russia). During this period, the patient displayed aggravation of PD clinical symptoms that were accompanied by a decrease in the ceruloplasmin concentration (from 0.33 to 0.27 g/L) and an increase in serum nonceruloplasmin copper, which are typical of the late stages of Wilson’s disease. It was found that one of the alleles of exon 14 in the *ATP7B* gene, which partially codes of the nucleotide-binding domain (N-domain), carries a mutation not previously reported corresponding to Cys1079Gly substitution. Alignment of the ATP7B N-domain amino acid sequences of representative vertebrate species has shown that the Cys at 1079 position is conserved throughout the evolution. Molecular dynamic analysis of a polypeptide with Cys1079Gly substitution showed that the mutation causes profound conformational changes in the N-domain, which could potentially lead to impairment of its functions. The role of *ATP7B* gene mutations in PD development is discussed.

## 1. Introduction

Parkinson’s disease (PD) is a widespread neurodegenerative disorder that affects more than 1% of people older than 60 and 4% older than 85, making it the second most prevalent neurodegenerative disorder after Alzheimer’s disease. The disease is characterized by a variety of motor (rest tremor, muscle rigidity, bradykinesia, postural instability) and nonmotor (cognitive problems, neuropsychiatric disturbances, gastrointestinal dysfunction, autonomic and sensory changes) symptoms, which are caused by the death of dopamine-secreting neurons in the *substantia nigra pars compacta* (SNpc) [1,2]. The accepted reason for this phenomenon is an aggregation of α-synuclein protein in Levi bodies, which are the characteristic attributes of PD [3]. Numerous reasons provoke α-synuclein aggregation with an oxidative stress being one of the most important among them [4,5], while itself oxidative stress may be a result of different processes, including the impairment of copper metabolism [6,7,8]. The biochemical manifestation of copper dyshomeostasis in PD patients is a decrease in the ceruloplasmin (Cp) protein concentration as well as Cp-associated oxidase activity [9,10], which correlates with the early onset of PD (EOPD) [11]. Cp is a blue multicopper (ferr)oxidase that belongs to moonlighting proteins. The main physiological functions of Cp consist of its roles in iron and neurotransmitter metabolism, and copper transport [12,13,14,15]. The well-proven function of Cp is radical oxygen species (ROS)-free oxidation from Fe(II) to Fe(III), while its impairment of this process leads to iron accumulation in the SNpc, which is also a typical trait of PD [16,17] and, therefore copper dyshomeostasis now is considered as a PD marker [18,19,20]. About 5–10% of PD cases have Mendelian inheritance [21,22]. There are about 20 genes with the mutations that are known to be responsible for disturbances in the cellular processes leading to neuronal death in SNpc, including mitochondrial dysfunction, defects in mitoautophagy and chaperone-mediated autophagy, defective dopamine metabolism, endoplasmic reticulum stress from protein aggregation, and mitochondrial calcium transport [23,24,25,26]. The remaining 90% of PD cases are typically classified as sporadic, which are believed to be caused by a combination of multiple etiological factors, including oxidative stress, and adverse and harmful environmental factors that are not fully understood [27]. A significant number of these cases may be patients with genetic predisposition, primarily heterozygous carriers of hereditary diseases that share symptoms with PD, e.g., Gaucher disease, Niemann–Pick disease, Wilson’s disease (WD), aceruloplasminemia, GM1 gangliosidosis, and some mitochondrial diseases [28,29,30,31,32].

In the present article, we report a case of PD with early onset in a patient who is a heterozygous mutation carrier of the *ATP7B* gene, which codes for copper-transporting ATPase P1 type, also known as ATP7B or Wilson’s ATPase [33]. The name of the ATPase is linked to WD, an autosomal recessive genetic disorder caused by mutations in the *ATP7B* gene [34]. The biochemical manifestations of WD are low Cp and copper levels in blood serum, excessive accumulation of toxic copper amounts in some organs (liver, brain, cornea), and excretion block through bile. These symptoms are caused by the fact that ATP7B provides copper atoms for Cp metalation and implements copper excretion [35]. In our study, we aimed to analyze clinical representations of PD during its progression in the heterozygous carrier of WD gene.

## 2. Materials and Methods

This study was approved by the Local Ethics Committee for Medical and Health Research at the Institute of Experimental Medicine, St. Petersburg, Russia (Protocol №2, 13.11.2012). In accordance with the Declaration of Helsinki, patients signed an informed consent form prior to participation in the study. Patient M. was clinically assessed using the Unified Parkinson’s Disease Rating Scale (UPDRS), Postural Instability and Gait Disorder (PIGD),Schwab and England Activities of Daily Living Scale (SE-ADL) [36], Hoehn and Yahr stage (H&Y) [37], Nonmotor Symptoms Scale for Parkinson’s disease (NMS PD) [38], Mini Mental State Examination (MMSE) [39], Beck’s Depression Inventor (BDI) [40], Hospital Anxiety and Depression Scale (HADS) [41], and Frontal Assessment Battery (FAB) [42]. Clinical grading was determined in the OFF state following the same protocol by at least two PD specialists.

Blood samples were collected from the cubital vein after overnight fasting and before medication intake. Serum was collected by centrifugation after clot formation at 5000× *g* for 10 min. Leukocytes were isolated from blood collected with anticoagulant (Sigma-Aldrich, St. Louis, MO, USA) by isopycnic centrifugation in 1.077 g/mL Ficoll-Hypaque (Sigma-Aldrich, St. Louis, MO, USA) [43]. Leukocyte chromosomal DNA was extracted by high-efficiency DNA extraction kits (AmpliSens Biotechnological, Moscow, Russia) according to the manufacturer’s protocols. Specific primers (F: tccatctgtattgtggtcag, R: cagctaggagagaaggacat) were used to amplify *ATP7B* exon 14, using the previously described PCR protocol [44]. PCR products were purified by DNA isolation kits (OMNIX, St. Petersburg, Russia). Sequencing was performed on a 3500 Series Genetic Analyzer (Applied Biosystems, Foster City, CA, USA). Cytokine concentrations were assessed by immunoassay kits, manufactured by VectrosBest (St. Petersburg, Russia). Oxidase-capable Cp concentration was evaluated in the reaction with *p*-phenylenediamine, as described earlier [45], and its protein concentration was determined by quantitative immunoelectrophoresis [46]. In the present work, noncommercial monovalent polyclonal rabbit IgG to highly pure human Cp (A_610/280_ = 0.050), obtained by the method proposed by Sokolov [47], were used. To precipitate Cp, 50 μL of blood serum were mixed with 100-fold excess of Cp antibodies. The mixture was incubated overnight at 4 °C; the precipitate was separated by centrifugation at 5000× *g* for 15 min, rinsed with PBS, and then dissolved in pure concentrated nitric acid. The copper concentration was measured in the solution. To bind labile copper ions, the serum was treated with Chelex 100 resin (BioRad, Solna, Sweden): 5 mg of the resin was added to 50 μL of the serum and incubated for 3 h at room temperature with constant stirring. The resin was then removed by centrifugation, and the supernatant was used for measure. Copper concentrations were measured on a graphite furnace atomic absorption spectrometer, ZEEnit 650P (Analytic Jena, Jena, Germany), with Zeeman correction of nonselective absorption [46].

Phylogenetic analysis was performed with MultAlin (http://bioinfo.genotoul.fr/multalin/multalin.html). The ATP7B N-domain experimental structure (1032–1196) was obtained from the Protein Data Bank (PDB ID: 2ARF, solution NMR). The first model from the available NMR models was used for the following in silico analysis. Based on the wild type (WT), the structures of the Cys1079Phe and Cys1079Gly were modeled in the Leap module of AMBER 16 (Assisted Model Building with Energy Refinement, University of California, San Francisco, CA, USA) [48]. The above-described structures were used as initial structures for the molecular dynamics (MD) analysis performed in the MD package AMBER 16 [48]. We used counterions (Na^+^) and periodic boundary conditions with a truncated octahedron TIP3P water box with at least 4 Å distance from the solute to the periodic box border, which corresponded to about ~10^4^ water molecules in the periodic box. Arg and Lys residues were protonated, while Asp and Glu residues were deprotonated in the simulations. His residues were protonated, which corresponded to the conditions (pH 6.0) under which the experiment structure was obtained [49]. An ff14SB force field was applied [50]. Two-step energy minimization was carried out: first, 0.5 × 10^3^ steepest-descent cycles and 10^3^ conjugate-gradient cycles with harmonic force restraints of 100 kcal/(mol·Å^2^) on solute atoms, and then 3 × 10^3^ steepest-descent cycles and 3 × 10^3^ conjugate-gradient cycles without restraints. Afterward, the system was heated up to 300 K for 10 ps with harmonic force restraints of 100 kcal/(mol·Å^2^) on solute atoms and equilibrated for 50 ps at 300 K and 10^5^ Pa in an isothermal isobaric ensemble (NPT). Finally, 10 ns of a productive MD run was carried out in an NTP ensemble. The SHAKE algorithm, 2 fs time-integration step, 8 Å cutoff for nonbonded interactions and the particle mesh Ewald method were used. We appended the coordinates to the trajectory file each 10 ps, making it 1000 frames in total. The last nanosecond of the MD simulation was used for postprocessing free-energy calculations performed by molecular mechanics‒generalized borne surface area (MM-GBSA) [51] using the modified GB model (igb = 2) [52]. Free energies were decomposed per residue and per residue pairs. Only enthalpic components were considered since entropic-component calculation remains a bottleneck of the MM-GBSA method and potentially increases the overall uncertainty substantially in the calculated free energies [53]. Trajectories were analyzed using the CPPTRAJ module of AMBER 16 and visualized in VMD [54].

## 3. Case Report

Patient M., a 57-year-old Caucasian male, revealed the first symptoms of PD at the age of 39, when “muscle contraction” of the left leg appeared during walking; in the following year, stiffness in the left hand also developed. The initiation of focal limb dystonia attracted specialist’s attention, as it is a typical symptom of EOPD, with onset at an age younger than 40. Patient M. displayed a slow rate of disease progression and a long period of exclusively unilateral hemiparkinsonism according to the clinical picture. In 2000, the neurologist of the outpatient clinic diagnosed PD, and the patient was initially started on levodopa/benserazide 100/25 mg thrice daily, which resulted in significant improvement of his condition. Levodopa dosage was gradually increased over the years. Patient M. has been observed in the clinic of the Institute of Experimental Medicine (IEM), St. Petersburg, since 2011. After 11 years of receiving oral medication, he experienced the “wearing-off” phenomenon. After 15 years of disease progression, a spontaneous deterioration in the patient’s condition was observed. In 2015, the “single-dose-depletion” phenomenon was accompanied by the wearing-off phenomenon; “peak-dose dyskinesia” also then appeared for the first time. During the whole time of observing the patient, akinetic‒rigid syndrome dominated over tremors. The basis of the diagnosis was clinical neurological manifestations, but the patient also had a typical PD reaction to levodopa, including the disappearance of hypokinesia, rigidity, tremors, and dystonia upon receiving levodopa, the development of fluctuations, and levodopa-induced dyskinesias. Currently, no case of WD with a positive effect from chronic levodopa therapy has been described. In confirmation of the PD diagnosis, hyposmia was also detected in patient M. To exclude a Kayser–Fleischer ring, a slit-lamp examination was performed by an experienced ophthalmologist. Patient M. underwent 3-Tesla magnetic resonance imaging (MRI) and no brain MRI abnormalities typical of WD were found. In addition, no hepatic representations, thrombocytopenia, or coagulopathy was identified. The patient was specifically asked about any previous episodes of jaundice or liver disease.

Cognitive function was evaluated by MMSE, and FAB was also employed for comparative analysis of cognitive disorders predominantly affecting the frontal lobes or the subcortex. PD stage was assessed with H&Y in Lindval modification. The severity of the disease was rated by UPDRS; specifically, UPDRS-I was used to rate the severity of nonmotor symptoms, UPDRS-II—activity in everyday life, UPDRS-II—motor symptoms, UPDRS-IV—frequency of complications of dopaminergic therapy. The manifestation of anxiety and depressive disorders was estimated by the 15-point scale of geriatric depression, the Sheehan Patient-Rated Anxiety Scale, the Beck Depression Inventory, and HADS “A” and “D”. Other nonmotor symptoms were rated by the PD-NMS scale; among them, constipation and sialorrhea were the most profound. There was no family history of movement disorders or cognitive dysfunction. The patient is employed and is not subjected to occupational hazards.

At observation in 2018, the following diagnosis was set: “Parkinson’s disease with early onset, akinetic-rigid type, stage 2.5 H&Y, predominantly left-sided lateralization of the symptoms, complicated with motor fluctuations: on–off phenomenon, wearing-off phenomenon, nonmotor fluctuations, chorea-like dyskinesia in drug-plateau period. Restless-legs syndrome.”

Full characteristics of the patient during his observation in the IEM clinic (from 2011 to 2018) by various diagnostic scales and checklists are given in Table 1. The motor disorders were observed against a relatively good neuropsychological background and a low severity of nonmotor symptoms, which is typical of EOPD patients.

Concentrations of inflammatory (IL1B, IL6, IL8, TNF) and anti-inflammatory (IL10) cytokines, as well as copper status indices, were measured in the blood serum of patient M. (Table 2).

In 2011, the levels of IL1B, IL6, and IL10 in blood serum were above the typical range of healthy people. At the second examination in 2018, the levels of these cytokines were not elevated. In 2018, the TNF and IL8 concentrations were higher than the reference values; this observation was not consistent with the literature data on the decrease of TNF and IL8 in PD patients [55]. It is worth noting that an increase in TNF and IL8, which is not typical of PD, was observed in WD patients, and it was found to be correlated with WD severity [56]. In the considered case, changes in the cytokine profile may be explained by progression of the disease, the appearance of complications caused by the pathogenic decrease of the number of neurons in SNpc by more than 90%, and/or modification of the patient treatment procedures during the seven years. The change of levodopa dose was the most significant change in the antiparkinson treatment scheme, as well as the introduction of other drugs affecting the level of inflammatory cytokines. However, it is hard to draw conclusions from the observed cytokine dynamics data in a single patient because IL and TNF levels in PD patients are highly variable and only future correlation analysis may reveal the relations [57,58].

Cp and copper concentration in the blood serum of patient M. decreased by ~20% during the time of observation (Figure 1). Meanwhile, the concentration of copper that was precipitated by antibodies to Cp became lower in 2018 (Figure 1A). If serum copper concentration is considered as 100%, then Cp-bound copper accounted for 87% of total copper in 2011, and for only 69% in 2018. In healthy individuals, Cp accounts for more than 95% of serum copper [59]. Thus, the quantity of serum copper that is not bound to Cp (non-Cp copper, also known as ‘free’ or exchangeable copper (CuEXC) [60]) increased in patient M. during the progression of the disease. Chelex 100 resin, which specifically binds labile cupric Cu(II) ions, bound less than 10% of the total serum copper. This indicated that, in the serum of patient M., non-Cp copper was generally unable to bind to Chelex 100. The concentrations of immunoreactive Cp protein and Cp oxidase activity decreased proportionally during the time of observations (Figure 1B–D). Together with an increase of non-Cp copper (Figure 1A) these phenomena are common WD features [60,61,62].

Summarizing, the obtained data suggested that patient M. may be a heterozygous carrier of WD. Taking into account that most WD patients in North America, Europe, and Russia possess mutations in exon 14 (e.g., E1064A, H1069Q, R1151H, and C1104F) coding for a part of the ATP7B nucleotide-binding domain (N-domain) of ATP7B (Figure 2A) [63,64], sequencing of this exon in patient M. was the most relevant option. We found a heterozygous genotype with substitution T/G at 3235 position in CDS, which corresponds to amino acid substitution Cys1079Gly (Figure 2B). The mutation was not found in general public database GnomAD. A similar but not identical mutation, Cys1079Phe, was described earlier in one of the chromosomes of a compound homozygote of a WD patient in China [65]. Alignment of the ATP7B N-domain protein sequences of representatives from all vertebrate classes (fishes, amphibians, reptiles, birds, and mammals) has shown that the Cys1079 residue is conserved (Figure 2C). Thirty PD patients from northwest Russia (the region of the Russian Federation with the highest frequency of PD) also took part in this investigation. None of them carried mutations in the C1079 position or other known mutations in *ATP7B* exon 14.

To understand how mutations in the ATP7B N-domain could influence its structure and, therefore, be reflected in the function of the protein, we performed MD-based WT analysis of it as well as of the Cys1079Phe and Cys1079Gly protein mutants. Per residue pair, free energy decomposition analysis clearly shows dramatic rearrangements in the protein region containing Cys1079 in the WT (Appendix A, Figure 3).

In particular, contact of the residue in position 1079 with Leu1057 and Lys1077 is weakened by both mutations; contact with Glu1082 is stabilized upon both mutations, with a more pronounced effect in Cys1079Phe, while contact with Tre1076 is substantially stabilized in Cys1079Gly; contacts with Val1060 and Leu1083 are also stabilized in both mutants. Visual inspection of the corresponding structures obtained in the MD simulation (Figure 4) suggests that the introduction of a bulkier Phe instead of Cys allows for establishing several additional interactions in Cys1079Phe, in which the π electrons of the phenyl ring of Phe1079 can directly interact with the Tyr1078 side-chain via the formation of an H-bond, while Arg1054 establishes π‒cation interaction (Figure 4C). The consequence of the latter could be a destabilization of the salt bridge formed between Arg1054 and Glu1082 in the WT, which, in turn, could locally affect the distribution of electrostatic potential, as well as the H-bonding propensity on the surface of the protein and/or its folding pathways. Therefore, Cys1079Gly mutation leads, in general, to a decrease in the number and strength of contacts (Figure 4D).

Since the introduction of a Gly residue within an α-helix is supposed to disrupt the folding of the helix [66], therefore potentially affecting the folding of the whole protein, in such a scenario the mutation can lead to a misfolded protein. However, because the performed MD simulation is too short and starts from an experimental structure corresponding to an energy minimum, this prevents direct observation of the possible misfolding. Probably due to the same reason, despite the clear rearrangements detected in the intraprotein interface, the individual impact of all the analyzed residues, which include the residue in position 1079 and its interacting counterparts, do not significantly differ between the WT and mutants (Appendix A). An analysis of the most important movements in the protein by the anisotropic network model approach [67] suggests that this protein region is very tightly packed and, therefore, not highly mobile. To summarize, our in silico description of the mutation effect proposes substantial rearrangements of the protein contacts that could have a crucial influence on the folding and, therefore, the function of the protein that could underlie distortions in the function of the protein resulting in WD. We identified the residues that, being in contact with Cys1079 in WT, are potentially important for the folding and/or function of ATP7B.

## 4. Discussion

The clinical features of patient M. correspond to typical PD according to the MDS criteria [2,68]. The onset of the disease and its progression were not accompanied by ataxia, chorea, dysarthria, dysphagia, or excessive salivation, or by behavioral abnormalities with alterations of personality, depression, psychosis that would be untypical of PD or would be indicative of WD. Hepatic symptoms were not observed either. The course of the disease was characterized by gradual aggravation of the severity; mostly motor symptoms were manifested, while normal cognitive status was preserved (Table 1). At the same time, copper status indices and their changes during disease progression (decline of Cp and copper concentrations, decrease in Cp-bound copper, and increase in non-Cp copper fractions in patient’s blood serum) corresponded to the typical ATP7B-dependent disturbance of the copper metabolism (Figure 1). The precise mechanism of these changes in WD has not yet been established. Still, it is known that within *Atp7b*−/− hepatocytes, copper is distributed nonuniformly, and this distribution changes as the disease progresses. In the early stages of WD, copper is preferentially elevated in the cytosol and nuclei [69]. Later, copper is redistributed to the mitochondria [70], and then appears in the bloodstream in a complex with a small copper carrier of unidentified nature [62], and with extracellular metallothionein [71]. It is interesting that the same copper redistribution is typical of the embryonic type of copper metabolism [46], which, in mammals, is a WD phenocopy. Both the small copper carrier and metallothionein bind copper ions in a Cu(I) oxidation state. In the blood serum, the major fraction of copper is bound to Cp in the Cu(II) state [72]. The Cp molecule contains six tightly bound copper atoms in the active centers that can be extracted under mild conditions, and 1–3 labile ions that can be removed by chelators. It is possible that we observed poor removal of copper by Chelex 100 (Figure 1) because it does not bind Cu(I), and the amount of labile Cu(II) bound to Cp was low [10].

Data on the influence of copper dyshomeostasis on the progression of PD are controversial. Indeed, it has been previously shown that the decrease of copper concentration in blood serum correlates with PD severity [10,73]. On the other hand, Kim et al. reported that the severity of the disease correlates with elevated copper concentrations [74]. Such controversies may have different reasons with the most important being that the total atomic copper concentration has poor biological relevance. This concentration is the sum of several copper pools with different roles: catalytic copper in Cp-active centers, labile Cu(II) associated with Cp, copper associated with albumin, α2-macroglobulin (transcuprein), and non-Cp ‘free’ copper with a not-fully-understood molecular environment. Changes in these pools may have different causes, such as the decrease in total copper concentration that is observed both in WD [34] and in impaired Cp expression (aceruloplasminemia). Heterozygous carriers of mutations in both genes (*ATP7B* and *Cp*) are linked to the development of PD [28,29,30]. However, in WD patients, the metalation of cuproenzymes and copper excretion through the bile are impaired. In aceruloplasminemia, deficiency in cuproenzymes and copper accumulation in the cells are not observed. Therefore, there may be no significant change in the total serum copper concentration. Still, the redistribution of copper between different pools can reflect severe impairments in copper homeostasis of various origins. The importance of factors responsible for copper dyshomeostasis (ecological factors, frequency of specific mutations) may also vary in different populations. The specificity of copper dyshomeostasis remains unaccounted for in studies of the relationship between PD and it may be the reason for the controversies in the previously reported data.

We described a novel mutation in the ATP7B N-domain leading to nonsynonymous substitution of cysteine by glycine Cys1079Gly (Figure 2). This substitution was not found in the gnomAD database (https://gnomad.broadinstitute.org). The score predictors of Cys/Gly substitution pathogenicity were 0.05 for the SIFT algorithm and 0.999 for the Poly-Phen2 algorithm, and CADD score for A3392C in ATP7B cDNA was 32, which means it belongs to the 0.1% of most deleterious mutations in the human genome. From molecular modeling, we can conclude that the studied substitution could potentially lead to changes in amino acid contacts in the domain (Figure 3 and Figure 4). It is known that widespread mutations in the ATP7B N-domain lead to reduced activity of ATP7B and WD development, and changes in the native conformation of the domain are among the important reasons for this event [75,76]. Our data allow us to propose that the Cys1079Gly mutation also leads to the deterioration of ATP7B N-domain function.

As many neurological symptoms are common in PD and WD patients, it was hypothesized that heterozygous carriers of the WD gene may increase the risk of PD development. The theoretical basis for the relationship between PD and heterozygous mutations in the WD gene was elucidated by S. Johnson and reflects the fact that most patients with PD have reduced ceruloplasmin concentrations, an intrinsic attribute of WD, and that copper dyshomeostasis has an impact on the maturation of brain-specific cuproenzymes. However, the role of the *ATP7B* gene in the development of PD has not been studied [77]. Cases of association of PD with mutations in the *ATP7B* gene are known. In Sardinia, a case of familial PD with very late manifestation in three sisters was registered [78]. The sisters did not display mutations in genes commonly associated with PD, i.e., *SNCA*, *PRKN*, *LRRK2*, or in the Huntington’s disease gene on chromosome 4. Molecular testing revealed a 15 bp deletion (–441/–427) in the 5′-UTR of one of the *ATP7B* alleles in all three sisters; this mutation is common in Sardinia, and was not observed in other populations [78,79,80]. The deleted region includes the part of the *cis*-regulatory element for Hepatic Nuclear Factor 3, which is responsible for the normal expression level of *ATP7B* [79]. In a cohort of 97 EOPD patients from Germany, a 65-year-old patient was found who carried an H1069Q mutation in an *ATP7B* allele [81]. The 52-year-old sister of this patient was also a heterozygous carrier of H1069Q mutation, but she did not display any signs of EOPD. In WD patients, homozygous or compound heterozygous H1069Q mutation may lead to WD at an early age, but may also manifest in septuagenarians [82]. These examples show that the heterozygous carriers of WD may develop PD with either early and late onset, and the onset may be provoked by environmental factors. The case presented here supports the idea that a heterozygous carrier of WD may phenotypically manifest as PD patients. In all described cases where PD was associated with a heterozygous mutation in WD gene, the mutant allele carried the most widespread mutations of the respective populations. This may be explained by the fact that PD patients with altered copper status indices are typically screened for most well-known mutations, while the full sequencing of the large chromosomal gene is considered to be too complicated and laborious. The use of such a strategy may lead to the underestimation of PD cases associated with *ATP7B* gene mutations.

Copper dyshomeostasis, manifested as an alteration of copper status indices, is not a mandatory feature of PD. Previously, we screened a cohort of 50 PD patients and only revealed 17 patients with copper status indices that were characteristic of heterozygous WD carriers [83], but our experience still indicates that copper status observation in PD patients is sensible. The timely detection of a heterozygous WD gene in patients with PD clinical symptoms and the prescription of chelation therapy may help to slow down or even prevent PD development.

Finally, the question may arise of whether identifying or testing WD genes in PD patients is economically viable. Traditionally, WD has been considered to be a rare genetic disorder with an approximate frequency of 30 individuals per 10^6^. However, in the United Kingdom, calculations based on DNA diagnostic data indicated that the frequency of individuals predicted to carry two mutant pathogenic *ATP7B* alleles was ~1 per 7000 individuals, with heterozygote mutations found in up to 2.5% of the general population. Moreover, the prevalence of WD is higher in Asian countries (58 per 10^6^) and in isolated communities (for example, the Canary Islands, 1 per 2600; Sardinia, 1 per 7000; [84] and references therein). These data prove that mutated WD genes are more frequently observed than is broadly accepted, and this serves as a motivation for more detailed research and analysis of heterozygous WD carriers among PD patients.

## Figures and Tables

**Figure 1 jpm-09-00041-f001:**
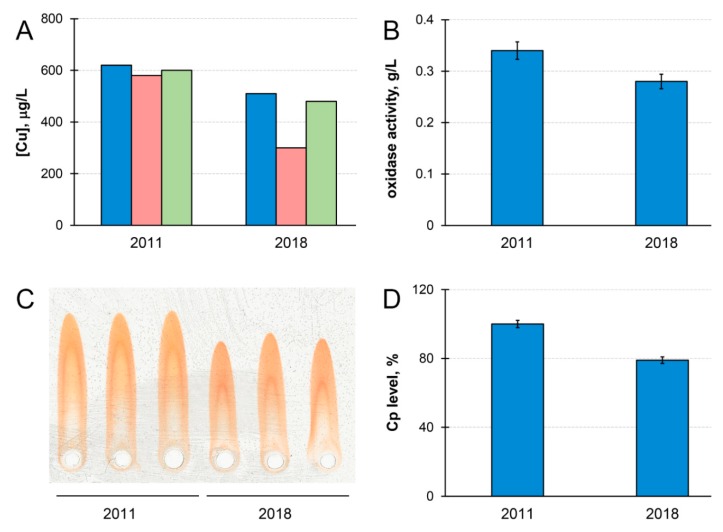
Chronological changes of copper status parameters in patient M.’s serum. (**A**) Serum atomic copper concentration (blue); copper concentration associated with ceruloplasmin (red); serum atomic copper concentration after Chelex 100 treatment (green) (µg/L). (**B**) Ceruloplasmin oxidase activity (g/L). (**C**) Immunoelectrophoregram of patient M.’s serum. From left to right: three replicas, 2011, and three replicas, 2018. (**D**) Diagram built according to immunoelectrophoresis data processing (%). Standard deviation (± SD) calculated from values of three independent measurements.

**Figure 2 jpm-09-00041-f002:**
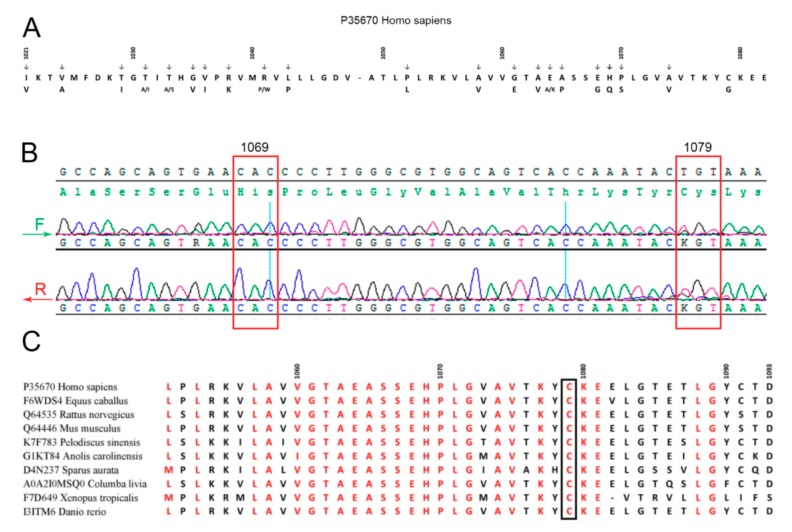
(**A**) Amino acid substitutions in the ATPase nucleotide-binding domain associated with Wilson’s disease. (**B**) Patient M.’s exon 14 sequencing. (**C**) Alignment of ATP7B vertebrate nucleotide-binding domains.

**Figure 3 jpm-09-00041-f003:**
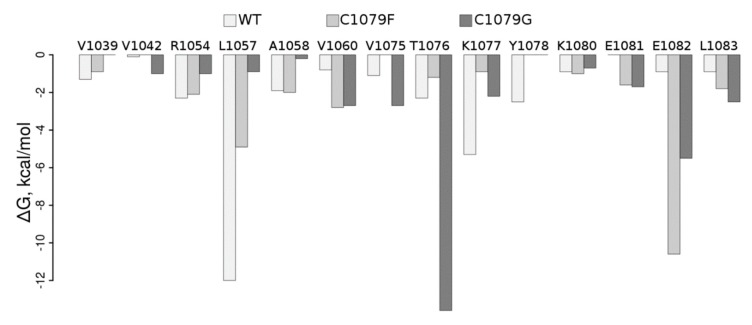
Molecular mechanics-generalized borne surface area (MM-GBSA) free energy per residue decomposition (ΔG, kcal/mol). Residues in the figure revealed the free energy of contact with the residue in position 1079 to be lower than 0.6 kcal/mol (~RT, T = 300 K) in at least one of the simulations.

**Figure 4 jpm-09-00041-f004:**
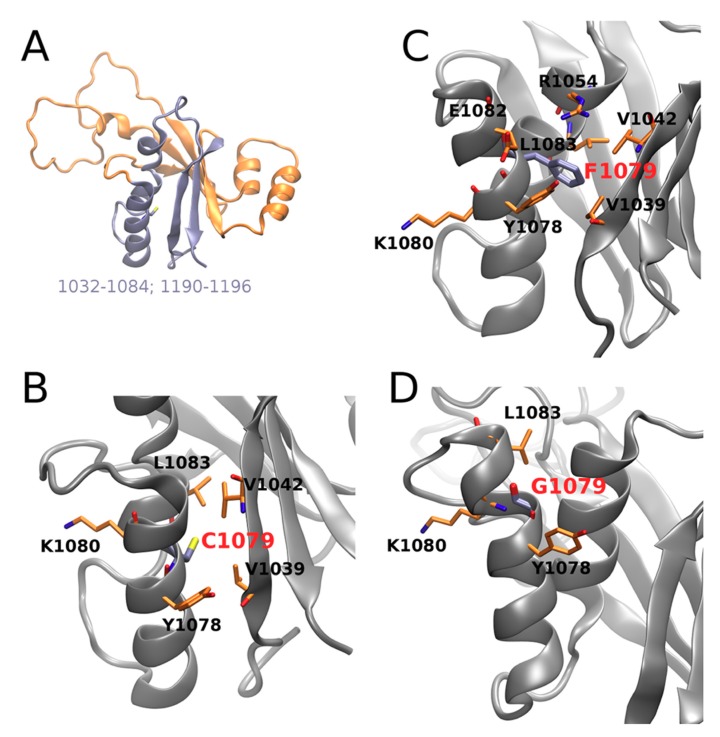
ATP7B N-domain (1032-1196, PDB ID: 2ARF, NMR the first model): (**A**) interface (in cyan) of C1079; (**B**–**D**) residue in position 1079 and its representative interaction residue counterparts in wild type, C1079F and C1079G, respectively. Protein backbones are shown as gray shapes, and particular residues as orange sticks.

**Table 1 jpm-09-00041-t001:** Parkinson’s disease clinical manifestation dynamics in patient M.

Parameters	Date of Observation		
	September 2011	October 2014	June 2018
Age (years)	51	54	57
Modified H&Y scale (stage)	1.5	2.5	2.5
UPDRS I (scores)	3	5	6
UPDRS II (scores)	7	10	11
UPDRS III (scores)	24	38	48
UPDRS IV (scores)	1	5	7
Total UPDRS (scores)	35	58	72
MMSE (scores)	30	28	28
FAB (scores)	18	18	18
Clock-drawing test (scores)	10	10	10
PD-NMS (scores)	13	26	32
BDI’s scale (scores)	15	14	17
Sheehan Clinical Anxiety Rating Scale (scores)	21	10	35
HADS «A» (scores)	5	4	5
HADS «D» (scores)	7	4	6

**Table 2 jpm-09-00041-t002:** Chronological shifts of the cytokine profile in patient M.’s blood serum.

Parameters (pg/mL)	Patient M.	
	September 2011	June 2018
Interleukin-1β	6.4	0
TNF-α	0	2
Interleukin-6	1.4	0
Interleukin-10	6	3
Interleukin-8	Not measured	14

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
