# Peer review of "Case of Early-Onset Parkinson’s Disease in a Heterozygous Mutation Carrier of the ATP7B Gene"

_jpm, 2019, doi:10.3390/jpm9030041_

Round 1
Reviewer 1 Report
This case report ist can be interesting for all physicians, especially for neurologists und may be important for the diagnostic of the Wilson's disease.
But, before publishing, this manuscript need some corrections:
Please, use in all manuscript instead of "Wilson disease" - Wilson's disease
Abstract:
Line 26: please add the exactly ceruloplasmin concentration
Keywords: please add also "Wilson's disease"
Introduction:
Line - 48-54: I miss the References, please include more References
Material and Methods:
Line 98: .. (Sigma-Aldrich, USA): please include also the city name, not only country name (USA). The same Line: 100, 104, 105, 114.
Line 109: please include the product information about this rabbit serum
Line 107: this reference is too old? 1970?
3. Case Report:
Line 149: which disease...?
Line 164: please include References after H&Y
The same LIne 167: after MMSE and FAB
The same Line 169,170, 171, 174,176: please include the References after all scales !!
4. Discussion
Line 280: please include the References after MDS criteria.
I am absolutely not sure that this case is Parkinson's disease!!
Please show me that this is really Parkinson's and not a typical primery Wilson's disease with neurological symptoms as usually. From my opinion, this case is more likely to be Wilson's disease with Parkinsonian symptoms.
Please, include more exactly grounds into Discussion.
What about the cornea of this patient and also about the Kayser-Fleische ring? What about another symptoms typical for PD and WD? Why is this case Parkinson's disease? Why not this case rather the WD with Parkinsonian symptoms? If you know, the patients with Wilson's disease shows a lot of different neurological symptoms, for example, very important two of them:
The majority of neurologic presentations consist of a movement disorder associated with bulbar symptoms.
The movement disorder is usually characterised by tremor, dystonia or parkinsonism.
Denny-Brown initially referred to two subgroups: one with predominant tremor referred and another with predominant dystonia, such as in this case. Please see a new literature:
Clinical presentations of Wilson disease
Samuel Shribman1, Thomas T. Warner1, James S. Dooley2
Submitted Mar 21, 2019. Accepted for publication Apr 09, 2019, doi: 10.21037/atm.2019.04.27
View this article at: http://dx.doi.org/10.21037/atm.2019.04.27
Please, include more exactly reasons and grounds, because in this case you think that is PD?
References
please do it according to Journal of Personalized Medicine!!!
Author Response
Dear Reviewer,
we are sincerely grateful to you for reviewing our article. All your comments have been considered. Please find our step-by-step answers in italics below. Please note that in the version, in which the Editorial Board asks for changes, the numbering of the lines differs from those indicated by you (submitted version). In the answers we followed the numbering of your version. And please accept our gratitude for participating in the improvement of our article.
Sincerely Your,
Authors
Reviewer: Please, use in all manuscript instead of "Wilson disease" - Wilson's disease
Authors: In the entire text "Wilson disease" is replaced by “Wilson’s disease”.
Reviewer: Line 26: please add the exactly ceruloplasmin concentration.
Authors: The data on ceruloplasmin concentration are added.
Reviewer: Line - 48-54: I miss the References, please include more References
Authors: the references are added.
Reviewer: Line 98: (Sigma-Aldrich, USA): please include also the city name, not only country name (USA). The same Line: 100, 104, 105, 114.
Authors: Now it’s Sigma-Aldrich, St. Louis, USA.
Reviewer: Line 109: please include the product information about this rabbit serum
Authors: In work, non-commercial monovalent polyclonal rabbit antibodies to high pure human Cp (A610/280 = 0.050) obtained by the method proposed by Sokolov [47] were used (Please, see lines 238-239, editable version).
Reviewer: Line 107: this reference is too old? 1970?
Authors: The measurement of Cp concentration CPU with para-phenylenediamine according to the method proposed by Sunderman & Nomoto is used without modifications at present. The article is cited positively in modern reviews addressing the problems of diagnosis of WD (Woimant F. et al. New tools for Wilson’s disease diagnosis: exchangeable copper fraction. Ann Transl Med. 2019; 7(Suppl 2): S70. doi: 10.21037/atm.2019.03.02). Therefore, we would like to keep this reference.
Reviewer: Line 149: which disease...?
Authors: Clarified – PD.
Reviewer: Line 164: please include References after H&Y
The same LIne 167: after MMSE and FAB
The same Line 169,170, 171, 174,176: please include the References after all scales!!
Authors: The references for all diagnostic scales from [36] to [42] are citied in methods.
Reviewer: Line 280: please include the References after MDS criteria.
Authors: the Reference “Postuma, R.B.; Berg, D.; Stern, M.; Poewe, W.; Olanow, C.W.; Oertel, W.; et al. MDS clinical diagnostic criteria for Parkinson's disease. Mov. Disord. 2015, 30, 1591-1601. doi: 10.1002/mds.26424.” is added.
Reviewer: I am absolutely not sure that this case is Parkinson's disease!!
Please show me that this is really Parkinson's and not a typical primery Wilson's disease with neurological symptoms as usually. From my opinion, this case is more likely to be Wilson's disease with Parkinsonian symptoms.
Please, include more exactly grounds into Discussion.
What about the cornea of this patient and also about the Kayser-Fleische ring? What about another symptoms typical for PD and WD? Why is this case Parkinson's disease? Why not this case rather the WD with Parkinsonian symptoms? If you know, the patients with Wilson's disease shows a lot of different neurological symptoms, for example, very important two of them: The majority of neurologic presentations consist of a movement disorder associated with bulbar symptoms.
The movement disorder is usually characterised by tremor, dystonia or parkinsonism.
Denny-Brown initially referred to two subgroups: one with predominant tremor referred and another with predominant dystonia, such as in this case. Please see a new literature:
Clinical presentations of Wilson disease Samuel Shribman1, Thomas T. Warner1, James S. Dooley2 Submitted Mar 21, 2019. Accepted for publication Apr 09, 2019, doi: 10.21037/atm.2019.04.27
View this article at: http://dx.doi.org/10.21037/atm.2019.04.27
Authors: The diagnosis of PD was established in accordance with the Movement Disorder Society (MDS) criteria. The total accuracy for the probable PD according to these criteria is more than 92%. It is shown that in the case of modeling multiple criteria diagnosis can be improved to the maximum predicted value of 98%, provided that the diagnosis is carried out by specialists in the field of movement disorders. The basis of diagnosis was not only clinical neurological manifestations but also typical for PD reaction to levodopa, including the disappearance of hypokinesia, rigidity, tremor and dystonia in patients receiving levodopa, the development of fluctuations and levodopa-induced dyskinesias. Currently, no case of WD with a positive effect from chronic levodopa therapy has been described. In confirmation of the diagnosis of PD, hyposmia was also detected in patient M. To exclude Kayser-Fleischer ring a slit lamp examination was performed by an experienced ophthalmologist. Patient M. was underwent 3-Tesla magnetic resonance imaging (MRI), no typical for WD brain MRI abnormalities were found. In addition, no hepatic presentations, thrombocytopenia or coagulopathy have also been identified. The patient was specifically asked about any previous episode jaundice or liver disease. Partially fragment included in the manuscript.
In our work, we considered the modern concepts of the WD clinic, and the references “Clinical presentations of Wilson disease” was added.
Reviewer 2 Report
In this article, Ilyechova E et al., describe an early onset Parkinson disease case carrying the heterozygous mutation Cys1079Gly in the ATP7B gene (a causal gene for Wilson disease). The authors rely on the hypothesis that heterozygous carriers of variants related to recessive hereditary diseases that share symptoms with PD, may increase the risk of PD development and progression. A detailed observational study is performed that provides comprehensive clinical information of the carrier and accurate in silico analyses proposing that the ATP7B Cys1079Gly mutation leads to the deterioration of the function at the ATP7B N-domain. The authors manifest that the case they present here supports the idea that heterozygous carriers of variants related to WD may manifest phenotypically like PD and therefore this should be further studied.
Although it is an interesting approach, I have the following concerns:
- Did this patient carry any mutations in genes commonly associated with PD?
- Do the authors have the DNA from the family members? Are the parents affected with PD and in that case, does this variant segregate with the disease?
- What is the frequency of this variant in the general public datasets (i.e GnomAD)? Is this a rare or common variant?
- What SIFT, Polyphen, CADD score or other predictors of pathogenicity tell us about this variant?
- Why was only exon 14 of the ATP7B gene screened and why this variant chosen?
- The manuscript is very long. It should have a concise and clear message. Maybe part of the less important content could be referred as part of the Supplementary information.
- The authors would benefit from a native English speaker
- All the gene names (α-synuclein referred to SNCA, PRKN, LRRK2…) should be in italics.
- Replace “harmful ecological factors” by “harmful environmental factors”
- In the title Atp7b should be in capital
Author Response
Dear Reviewer,
we are sincerely grateful to you for reviewing our article. All your comments have been considered. They were very useful to us and we hope that we have managed to improve the manuscript. Thank you very much again. Please find our word-to-word answers in italics below.
Sincerely, Authors
Reviewer: What is the frequency of this variant in the general public datasets (i.e GnomAD)? Is this a rare or common variant?
Authors: We did search this genetic variant in GnomAD, when patient M’s exon 14 was sequenced by us. You are right that this should be in the text. Now added.
Reviewer: What SIFT, Polyphen, CADD score or other predictors of pathogenicity tell us about this variant?
Authors: According to SIFT: Substitution at pos 48 from C to G is predicted to AFFECT PROTEIN FUNCTION with a score of 0.05; to Poly-Phen2: this mutation is predicted to be PROBABLY DAMAGING with a score of 0.999, and CADD score for A3392C in ATP7B cDNA is 32 which means it belongs to 0.1% most deleterious mutations in human genome (less than 0.1% of all possible nutation are predicted to be worse than this one). We added these data in Manuscript.
Reviewer: Why was only exon 14 of the ATP7B gene screened and why this variant chosen?
Authors: It was possible to completely sequence the ATP7B gene of patient M. However, in spite of the fact that this is a difficult technical task for such a large gene, it can be expected that the putative mutation is located in a remote promoter region (i.e., remains not captured), or in the intron (i.e., in silico it will not be possible to verify its functional role), and even be a trans-mutation. While exon 14 contains the most common mutations associated with WD in the Russian population. Therefore, in our circumstances, it was advisable to start with exon 14. After this new mutation, not represented in the GnomAD database, was detected in a heterozygous state, we did not look for other mutations, considering that if patient M. were a compound homozygous, it would very likely have developed Wilson's disease.
Reviewer: The manuscript is very long. It should have a concise and clear message. Maybe part of the less important content could be referred as part of the Supplementary information.
Authors: We have already placed 2 tables in Supplementary Materials and would like to keep the manuscript in its current form.
Reviewer: The authors would benefit from a native English speaker
Authors: We did.
Reviewer: All the gene names (α-synuclein referred to SNCA, PRKN, LRRK2…) should be in italics.
Authors: Thank you, we did it.
Reviewer: Replace “harmful ecological factors” by “harmful environmental factors”
Authors: We did it.
Reviewer: In the title Atp7b should be in capital
Authors: We did it.
Round 2
Reviewer 2 Report
Dear authors,
My comments have not been entirely addressed. Please, address them as follows:
- Introduction:
“There are about 20 genes, the mutations in which are known to be are responsible for the disturbances in the cellular processings leading to neuronal death in SNpc”
Replace Processings by processess
What genes? Could you please add a reference of a PD genetics review here?
- Introduction:
"mitochondrial dysfunction, defects in mitoautophagy and chaperone-mediated autophagy, defective dopamine metabolism, endoplasmic reticulum stress etc"
What does etc mean?
- ATP7B gene continues not to be in italics. For instance:
Abstract: “which is typical for the late stages of Wilson’s disease. It was found that one of the alleles of exon 14 in ATP7B gene”
Discussion:
“DNA-diagnostic data allowed to consider that 422 frequency of individuals predicted to carry two mutants pathogenic ATP7B alleles”
“The use of such strategy may lead to the underestimation of 410 the PD cases, associated with ATP7B mutations”
- Please, the authors need a native speaker. For instance, these expressions are incorrect:
“In our study we aimed was to analyze clinical 89 representations of PD during its progression in a the heterozygous carrier of WD gene”
“Caucasian ethnic group, revealed the first symptoms of the PD “
“it was hypothesized that heterozygous carriers of WD gene may increase a risk of PD development”
- Please replace ecological by environmental in the following expression:
“Develop PD with both early and late onset, and the onset may be probably provoked by ecological factors”
Author Response
Dear Reviewer,
we are sincerely grateful to you for reviewing our article. Your comments have been considered. Please, find our step-by-step answers in italics below. Please note that this version is edited by MDPI English Editing.
Reviewer: My comments have not been entirely addressed. Please, address them as follows:
- Introduction:
“There are about 20 genes, the mutations in which are known to be are responsible for the disturbances in the cellular processings leading to neuronal death in SNpc”
Replace Processings by processes
What genes? Could you please add a reference of a PD genetics review here?
Authors: The word “processings” is replaced by “processes”.
In the shot Introduction, we list the genes whose responsibility for the development of PD is well documented. Since the article is not a review, we do not consider addition of new references necessary.
Reviewer: "mitochondrial dysfunction, defects in mitoautophagy and chaperone-mediated autophagy, defective dopamine metabolism, endoplasmic reticulum stress etc"
What does etc mean?
Authors: etc ois replaced by other and “from protein aggregation, mitochondrial calcium transport and possible other” is added.
Reviewer: ATP7B gene continues not to be in italics. For instance:
Abstract: “which is typical for the late stages of Wilson’s disease. It was found that one of the alleles of exon 14 in ATP7B gene”
Authors: This blooper is corrected. Thank you very much.
Reviewer: Discussion:
“DNA-diagnostic data allowed to consider that 422 frequency of individuals predicted to carry two mutants pathogenic ATP7B alleles”
“The use of such strategy may lead to the underestimation of 410 the PD cases, associated with ATP7B mutations”
“In our study we aimed was to analyze clinical 89 representations of PD during its progression in a the heterozygous carrier of WD gene”
“Caucasian ethnic group, revealed the first symptoms of the PD “
“it was hypothesized that heterozygous carriers of WD gene may increase a risk of PD development”
Authors: Unfortunately, in our article there is no exact correspondence to the indicated expressions. Also, the numbers functions (422, 410, 89) are unknown. Now the manuscript is checked by MDPI English Editor.
Reviewer: Please replace ecological by environmental in the following expression: “Develop PD with both early and late onset, and the onset may be probably provoked by ecological factors”.
Authors: Replaced.